# Diversified Rice Farms with Vegetable Plots and Flower Strips Are Associated with Fewer Pesticide Applications in the Philippines

**DOI:** 10.3390/insects14100778

**Published:** 2023-09-22

**Authors:** Finbarr G. Horgan, Enrique A. Mundaca, Buyung A. R. Hadi, Eduardo Crisol-Martínez

**Affiliations:** 1EcoLaVerna Integral Restoration Ecology, Bridestown, Kildinan, T56 P499 County Cork, Ireland; ecrisol@coexphal.es; 2Centre for Pesticide Suicide Prevention, University/BHF Centre for Cardiovascular Science, University of Edinburgh, Edinburgh EH16 4TJ, UK; 3School of Agronomy, Faculty of Agrarian and Forest Sciences, Catholic University of Maule, Casilla 7-D, Curicó 3349001, Chile; emundaca@ucm.cl; 4International Rice Research Institute, Makati 1226, Metro Manila, Philippines; buyung.hadi@fao.org; 5Plant Production and Protection Division, Food and Agriculture Organization of the United Nations, Vialle delle Terme di Caracalla, 00153 Rome, Italy; 6COEXPHAL (Association of Vegetable and Fruit Growers of Almeria), Carretera de Ronda 11, 04004 Almeria, Spain

**Keywords:** biological control, biopesticides, ducks, sustainable farming, *Trichogramma*, wild harvesting

## Abstract

**Simple Summary:**

Smallholder farmers in Southeast Asia produce rice in flooded plots (<2 ha) surrounded by raised levees (bunds). To decrease pesticide use among farmers, researchers have promoted ecological engineering as a series of practices that optimize farm diversification to enhance the activities of pests’ natural enemies and reduce pest damage. This study examined the impact of farm diversification and other sustainability practices on pesticide use by rice farmers in the Philippines. We interviewed 302 farmers to assess their farm management practices. Many of the farmers used upland areas and bunds to produce fruits and vegetables. Some made botanical extracts of chili (*Capsicum* spp.), ginger (*Zingiber officinale* Roscoe), or lemongrass (*Cymbopogon* sp.) to control pests and diseases in their vegetables. In one region, the farmers avoided insecticides by using *Trichogramma* wasps to control stemborers. We found that farmers with relatively high awareness of the beneficial insects that occurred on their farms, who raised ducks in their rice fields, or who planted flowers or vegetables on their bunds tended to perform fewer pesticide applications to their rice crops. We recommend that flower and vegetable strips be combined with a series of other, environmentally friendly pest management options to enhance the outcomes of ecological engineering on rice farms.

**Abstract:**

Ecological engineering is defined as the design of sustainable ecosystems for the benefit of both human society and the environment. In Southeast Asia, researchers have applied ecological engineering by diversifying farms using flower strips to restore regulatory services to rice ecosystems and thereby reduce herbivore-related yield losses and overall pesticide use. We conducted a survey of 302 rice farmers across four regions of the Philippines to assess their farm diversification practices and determine possible associations with pesticide use. Rice was the main product on all farms; however, the farmers also produced fruits and vegetables, either rotated with rice (47% of the farmers) or in small plots in adjacent farmland. In addition, 64% of the farmers produced flowers, herbs, and/or vegetables on rice bunds. Vegetables were cultivated mainly to supplement household food or incomes, but 30% of the farmers also believed that the vegetables reduced pest and weed damage to their rice. We found that 16% of the farmers grew flowers on their bunds to reduce pest damage to rice and vegetables, and many farmers applied botanical extracts, growth stimulants, and insect traps to reduce damage to the vegetables. Some farmers avoided insecticides on rice by using *Trichogramma* cards. Planting flowers on rice bunds, rearing ducks in the rice fields, and farmers’ recognition of beneficial rice arthropods were statistically significantly associated with lower pesticide (particularly, insecticide) applications to rice. Our results indicate that farm diversification to produce supplementary foods for rural households and access to alternative pest management options can reduce pesticide use on rice farms in tropical Asia.

## 1. Introduction

Rice is the main staple food for over half of the world’s human population [1,2]. Much of rice production occurs in the tropical coastal lowlands of South and Southeast Asia [1,3]. Rapid population growth (i.e., the global population is estimated to exceed 9 billion by 2050), particularly in tropical Asia, has increased pressures on Asian farmers to intensify rice production [2,4]. In response, rice intensification practices including the use of high-yielding rice varieties, increased mechanization, and an increasing use of agrochemical inputs, have been promoted by national and international agricultural development institutes, often in partnership with the private sector [4,5]. Pesticide use, in particular, has increased dramatically among Asian farmers in recent decades: this is partly due to massive increases in global chemical production since the beginning of the millennium and intense marketing by agrochemical companies [6,7,8,9]. A high use of chemical pesticides at large scales and the co-dependence of certain technologies (i.e., hybrid rice varieties and direct seeded rice are associated with higher pesticide use than traditional varieties and establishment methods [10]) has resulted in a technological lock-in with respect to pesticide use, whereby the increasing use of pesticides reduces farmers’ willingness to adopt more environmentally friendly pest and weed management options [11,12,13].

In well-managed rice fields, a diversity of generalist and specialist natural enemies, a high degree of intraguild predation, and a range of interconnected negative feedback loops regulate arthropod densities such that rice herbivores normally occur in relatively low numbers and decline in abundance as the crop matures [14,15]. However, natural enemies are often highly vulnerable to insecticides and other pesticides [16]. Indeed, outbreaks of key rice pests such as planthoppers, stemborers, and leaffolders have been linked to excessive pesticide use throughout Asia [16,17,18]. These outbreaks were associated with a declining abundance of natural enemies, particularly during early rice crop stages [14,16]. In response to insecticide-related perturbations of rice arthropod communities and the consequent widespread and largescale outbreaks of rice pests, researchers proposed that farmers should avoid resurgence-causing insecticides and reduce overall insecticide use [16,19]. For example, largely in response to severe outbreaks of planthoppers in Thailand between 2009 and 2011, the Thai government, with support from the Thai Agro-Business Association (TABA), campaigned against the use of abamectin and cypermethrin in rice [20]. A range of rice sustainability programs throughout Southeast Asia have also called on farmers to limit pesticide use, particularly during early crop stages when natural enemies must build up their numbers in rice fields [21,22]. Furthermore, agricultural research and extension services throughout Asia continue to promote integrated pest management (IPM) and the adoption of alternatives to pesticide use such as community-based biological control, synchronized rice planting, and the use of resistant or tolerant rice varieties [10].

Because the large-scale use of pesticides over many years could be linked to a reduction in the diversity and abundance of natural enemies, several researchers have promoted the diversification of Asian rice landscapes using ecological engineering approaches to provide habitat and refuges for natural enemies and to restore regulatory ecosystem services [19,22,23]. Ecological engineering is defined as the design of ecosystems using engineering principals to promote benefits for both human societies and the environment [24]. In crop production systems, ecological engineering often relies on the use of functional plants (e.g., trap plants, repellent plants, or plants that provide alternative food sources for natural enemies) to increase the diversity and abundance of predatory arthropods. For example, the use of *Lobularia maritima* (L.) Desv. as a selective food plant for *Trichogramma carverae* Oatman and Pinto improved the biological control of *Epiphyas postvittana* (Walker) in Australian vineyards [25]. In rice ecosystems, farmers are encouraged to plant strips of vegetation on bunds as a key ecological engineering practice [19,26,27,28,29]. Reports from China, Thailand, Vietnam, Cambodia, the Philippines, and Bangladesh have shown that flower strips and planted bunds can increase natural enemy diversity and abundance, increase the mortality of rice herbivores due to egg parasitoids, mirid bugs, and spiders, thereby reducing pest densities, and bring added benefits to rice production systems such as increasing the diversity of insectivorous and snail-eating riceland birds, providing additional farm products for home use or sale to markets, and improving farm aesthetics [19,22,23,26,30,31].

In 2013, the Philippine Government (Department of Agriculture—Bureau of Agricultural Research (DA-BAR)) initiated a program to develop and promote ecological engineering as a means to diversify rice production systems and reduce insecticide use among rice farmers. This initiative would complement ongoing activities in the country to improve rice farm productivity through diversification [32,33] and to meet the proposed biodiversity targets for sustainable rice systems as set out in the country’s National Biodiversity Strategy and Action Plan [34]. As part of the program, demonstration rice farms were established in four rice-producing provinces (Laguna, Rizal, Iloilo, and Bukidnon) to promote ecological engineering among DA staff and local farmers [30]. However, even before initiating the DA-BAR program, it was apparent that many Filipino rice farmers already planted flower or vegetable strips on their rice bunds (levees) without any formal knowledge of the principles and practices of ecological engineering [33,35]. In this context, we conducted surveys of farmers in the four regions to establish a baseline for monitoring and to determine existing crop and pest management actions that affected pesticide use by the rice farmers. We also assessed whether the farmers’ recognition of the functions of rice arthropods was associated with pesticide inputs in rice and other crops and whether the farmers’ adoption of sustainable pest management practices such as the use of biological control agents, crop rotations, the rearing of livestock, or the planting of flowers on bunds, was associated with reduced pesticide use. Based on our results, we provide a series of recommendations to adapt ecological engineering to the existing practices for sustainable pest management used by farmers in the targeted regions.

## 2. Materials and Methods

Surveys were conducted in four provinces on three islands of the Philippines (Appendix A). Although the sites were in different provinces, our sampling was not sufficient to compare trends at a provincial level; therefore, we henceforth refer to each of our sites as located in one of four ‘regions’. The survey sites were selected based on proximity to established ecological engineering demonstration farms. These farms were described in detail by Horgan et al. (2017) [30] and were not yet seen by the farmers at the time of the interviews. The interviews were conducted at centralized locations (e.g., schools, village halls, or DA facilities) with farmers from adjacent towns and villages invited by DA staff through village leaders. 

Two of the regions are located on Luzon Island: in Rizal Province, farmers from 14 villages were interviewed at Pililia and Tanay, while in Laguna Province, farmers from 22 villages were interviewed at Victoria, Pila, and Nagcarlan. In both regions, the villages are mainly located on lowlands around Laguna de Bay, with Rizal to the northwest and Laguna to the south (Appendix A). The farmers, located near the lake and with access to irrigation infrastructure, generally produce two rice crops per year. On Panay Island, farmers from 16 villages were interviewed in Iloilo Province at Zarraga, Dumangas, and Dingle. These villages are located east of Iloilo City (Appendix A); the farmers mainly plant rice in rainfed paddies; therefore, many of the farmers lack sufficient water to produce rice during the dry season. Finally, on Mindanao Island, farmers from 21 villages in Bukidnon Province were interviewed at a school hall in Malaybalay. Bukidnon is an upland plateau with ample irrigation due to its mountainous terrain and proximity to the Pulangi River. Farmers in the region produce two rice crops per year, but many farmers also produce fruits and vegetables for local and export markets. 

### 2.1. Farmer Surveys

Farmers at each of the four sites were interviewed using a structured and standardized questionnaire. In total, 302 farmers were interviewed (56 in Rizal, 56 in Laguna, 96 in Iloilo, and 94 in Bukidnon). To facilitate the interviews, the questionnaire was translated from English into three local languages, i.e., Tagalog for Rizal and Laguna, Ilonggo for Iloilo, and Cebuano for Bukidnon. The translations were conducted by native speakers of each language with a knowledge of rice farming and pest management. The interviewers were DA staff at each centralized location and staff from the International Rice Research Institute (IRRI) based in Los Baños, Laguna. The interviewers were trained in interview methods, and each interviewer was familiar with rice production systems and crop management. A total of 61 interviewers conducted the one-to-one interviews. This relatively large number of interviewers was required because of the different languages used in the provinces. 

The interviews were conducted in two parts. The first part was conducted at the beginning of the day, and the second one toward the end of the day after the farmers had been shown the ecologically engineered demonstration fields (i.e., pre- and post-field event surveys). The organizers of the field events and surveys in each province were instructed not to disclose the topic of the interviews to the village leaders before the pre-field event interviews took place. This study used data collected only during the pre-field event surveys. A future paper will describe the field events and farmer responses to the events.

The questionnaire was developed according to the knowledge, attitudes, and practices survey technique (KAP) [36]. This technique is relatively robust and is resilient to varying interviewer experience. The questionnaire was developed based on Focus Group Discussions (FGDs) with farmers, seed suppliers, and DA staff at Laguna and Iloilo and on pre-testing in Laguna. The responses during the FGDs were used to code the questionnaires and, thereby, facilitate both the interview process and translations prior to data entry. However, we incorporated triangulation, and the interviewers were encouraged to also record qualitative information to aid in cross-checking the farmers’ responses. 

The final questionnaire consisted of four main sections, i.e., (1) farmer profiles, (2) farmers’ production constraints and pest and weed management practices in their main rice fields, (3) other crops produced on farms and related pest management approaches by farmers, and (4) farmers’ management of rice bunds. As part of Section 2, the farmers were shown photographs of eight rice arthropods and asked whether they considered each as a pest or as a beneficial organism on the farm. These arthropods were a dragonfly (*Agriocnemis pygmea* (Rambur)), a native bee (*Amigilla* sp.), a rice planthopper (*Sogatella furcifera* (Horváth)), a vespid wasp (*Vespa* sp.), a rice bug (*Leptocorisa oratoria* (Fabricius)), a mirid bug (*Cyrtorhinus lividipennis* Reuter), a ladybeetle (*Micraspis crocea* (Mulsant)), and a parasitoid wasp (*Anagrus* sp.). 

Prior to conducting the interviews, the interviewers informed each farmer about the objectives of the interview, how the data would be used, and how the data would be stored (including that the farmers’ names would only be recorded to match pre- and post-field day interviews, after which the names would be deleted, such that the reported results could not be linked to individual farmers). The farmers were also advised that they were not obliged to answer any questions. 

### 2.2. Data Analyses

The farmers’ responses were analyzed with region as the main factor. Nominal and ordinal variables (e.g., apply pesticides or not, rotate crops or not, grow vegetables or not, raise ducks, harvest snails, etc.) were analyzed using χ^2^ tests. Categorical variables were initially analyzed using 2 × 2 contingency tables or log-likelihood ratios (L-R χ^2^) to compare frequencies within categories (e.g., educational levels attained, reasons for avoiding pesticides, pest management actions, etc.). Contingency tables were also used to compare perceived biotic constraints. Tests of homogeneity and of mutual and partial independence were then conducted for any significant associations using χ^2^ analyses. Farmers’ recognition of rice field arthropods as beneficial or pestiferous was initially analyzed using χ^2^ tests. 

A simple index of the farmers’ abilities to recognize rice arthropods was calculated as the sum of the species that each farmer correctly assigned to its corresponding category from the eight images shown during the interviews. Because we included more beneficial arthropods than pest arthropods in the test and because most farmers recognized the two pest species (see below), this score also approximated the farmers’ ‘*leniency*’ toward rice field arthropods. We refer to this index as the farmers’ arthropod recognition scores.

Continuous dependent variables (e.g., farmers’ ages, experience in rice farming, numbers of applications of different pesticides, applications by farmers to rice or vegetables, arthropod recognition scores, etc.) were analyzed using univariate general linear models (GLMs). We used Tukey post-hoc tests to assess homogenous farmer categories. Residuals were examined to verify normality and homogeneity. 

Distance-based linear models (DistLMs) [37] were used to identify which variables best predicted pesticide use. Five models were used, one for each dependent variable, i.e., insecticide use, herbicide use, fungicide use, molluscicide use, and pesticide (i.e., the sum of all the latter) use. A total of 20 predictor variables were included in each DistLM analysis (Appendix A). Spearman’s correlation tests were used to check potential correlations between all predictor variables; none were found (R^2^ ≤ 0.7). Each DistLM was run using a stepwise routine, based on the lowest AICc (Akaike’s Information Criterion corrected) selection criterion, with 999 permutations. The similarity percentages routine (SIMPER) was used to analyze the differences in the farmers’ perceptions of the most harmful pests for rice production in wet and dry seasons. SIMPER analyses estimated which pests contributed most to generate dissimilarities (i.e., differences in perception) between pairs of regions [37]. The cut-off dissimilarity percentage was set at a minimum of 50% between pairs of regions. DistLM and SIMPER analyses were performed using the PRIMER V6 statistical package with the PERMANOVA+ add-on (PRIMER-E Ltd., Plymouth, UK).

## 3. Results

### 3.1. Farmer Profiles

The farmers we interviewed were on average > 50 years old and were predominantly male (except at Iloilo) (Appendix A). The farmers at Iloilo had also attained a higher level of education that those in Laguna or Rizal. Over 80% of the farmers at all sites produced rice as their main source of income; the farmers had on average > 20 years of rice-farming experience. The farms tended to be larger at Bukidnon (3.02 ± 0.28 ha vs. 1.52 ± 0.17 ha at Laguna, 1.84 ± 0.33 ha at Rizal, and 1.65 ± 0.13 ha at Iloilo), with a greater proportion of farmers owning their own land (94.27 ± 1.94% at Bukidnon, vs. 36.96 ± 6.59% at Laguna, 24.89 ± 5.64% at Rizal, and 64.80 ± 4.79% at Iloilo, Appendix A). More Bukidnon farmers also tended to produce other crops for income (35.11% vs. 7.14% at Laguna, 13.79% at Rizal, and 18.75% at Iloilo) but were less likely to own shops or other retail businesses compared to farmers at the other sites (5.32% vs. 17.86% at Laguna, 15.52% at Rizal, and 12.50% at Iloilo) (Appendix A). Nevertheless, a large proportion of farmers at all sites (44–75%) produced some other crops on their farms—often for home consumption. On average, 11–19% of the total farm area was dedicated to crops other than rice. For further details, see Appendix A.

### 3.2. Farmers’ Perceptions of Biotic Constraints in Rice

The farmers reported a range of constraints affecting their rice yields; these differed between sites (L-R χ^2^ = 237.363, DF = 96, *p* < 0.001) and between wet and dry seasons (L-R χ^2^ = 55.364, DF = 32, *p* = 0.006) (Appendix A). However, the differences in the farmers’ perceptions of biotic constraints to rice production in the four regions were mainly (>50%) determined by five pests (stemborers (*Chilo* spp.; *Scirpophaga* spp.; *Sesamia* sp.), ricebugs (*Leptocorisa* spp.), black bugs (*Scotinophara* spp.), rats, and golden apple snail (*Pomacea canaliculata* Lamarck)), irrespective of the season (see the results of the SIMPER analyses in Appendix A). The rank order of the most harmful pests was similar between seasons, except for rats, which were considered more harmful during the wet season compared to the dry season (Appendix A). The farmers in Bukidnon considered stemborers and black bugs as more harmful pests during both seasons than the farmers at the other sites (Appendix A). Rice bugs and rats were considered more harmful in Rizal than in the other regions during the dry season, but during the wet season, the farmers from both Laguna and Rizal perceived both rice bugs and rats as problematic. Apple snails were perceived as problematic by the farmers from Laguna during the dry season and by those from Iloilo during the wet season (Appendix A). Weeds and rice diseases were not regarded as serious problems by the farmers at any of the sites (mentioned <6% of the time as top-ranking constraints, Appendix A).

### 3.3. Farmers’ Recognition of Beneficial Rice Field Arthropods

The farmers generally recognized dragonflies and bees as beneficial insects, but were less likely to recognize mirid bugs, ladybeetles or parasitoid wasps as beneficial (Table 1). Many farmers, but particularly the farmers at Rizal and Iloilo, categorized ladybeetles (41–81%), parasitoid wasps (31–62%), and vespid wasps (12–46%) as pests. Although not statistically significant, there was a tendency for the farmers at Rizal (74%) to also categorize mirid bugs as pests, although more that 50% of the farmers at the other sites also felt that mirid bugs were pestiferous. A high proportion of farmers at all sites categorized planthoppers (>85%) and rice bugs (>90%) as pests (Table 1). Overall, the farmers at Laguna (arthropod recognition score = 6.02) and Bukidnon (6.02) correctly categorized insects as beneficial or pestiferous more times than the farmers at Rizal (4.88) and Iloilo (5.23) (F_3,297_ = 10.636, *p* < 0.001).

### 3.4. Rice Pest Management

The farmers performed an average of 1.10 herbicide applications, 1.08 fungicide applications, 1.67 insecticide applications, and 0.91 molluscicide applications to their rice crops per season (Figure 1A). More insecticide applications were made per crop at Rizal and Iloilo (F_3,293_ = 8.437, *p* < 0.001), and more molluscicide applications were made at Iloilo than at the other sites (F_3,293_ = 8.437, *p* < 0.001) (Figure 1A). The main insecticides used were cypermethrin, lambda cyhalothrin, chlorpyrifos, β-cypermethrin, and cartap hydrochloride. The farmers used metaldehyde and niclosamide to control apple snails in their rice (only two farmers used a saponin-based molluscicide). The farmers performed similar numbers of herbicide (F_3,294_ = 0.697, *p* = 0.555) and fungicide (F_3,294_ = 1.130, *p* = 0.337) applications across the regions. The main herbicides used were butachlor, 2-4D, benzobicyclon, pretilachlor, and fenoxaprop-p-ethyl. The main fungicides used were copper hydroxide, difenoconazole + propiconazole, cuprous oxide, benomyl, and mancozeb. A small number of farmers at each site also used carbofuran, possibly to control nematodes. 

On average, 12–34% of the farmers applied no insecticides, with more insecticide-free farmers at Laguna and Bukidnon (χ^2^ = 11.427, DF = 3, *p* = 0.010, Figure 1B). Although the farmers made a similar number of fungicide applications, more farmers at Iloilo (23%) avoided fungicides compared to other regions (41–51%) (χ^2^ = 17.705, DF = 3, *p* = 0.001: Figure 1B). But the farmers at Iloilo made more overall pesticide applications (5.71 ± 0.27) than those in the other regions (Laguna = 4.26 ± 0.26; Rizal = 4.90 ± 0.39; Bukidnon = 4.46 ± 0.32: F_3,294_ = 5.144, *p* = 0.002). Over 15% of the farmers at Bukidnon applied no pesticides, whereas <10% of the farmers at the other sites applied no pesticides (χ^2^ = 10.176, *p* = 0.008).

### 3.5. Farmers’ Reasons to Avoid Pesticides in Rice

The reasons why the farmers did not use insecticides varied between the sites (Table 2: L-R χ^2^ = 30.443, DF = 18. *p* = 0.033). At Bukidnon, the farmers mainly avoided pesticides by using alternative pest management options (69%), whereas at the remaining sites, the farmers believed that insects were not a problem (29–35%) or that natural enemies sufficiently controlled pests (20–43%). Nevertheless, between 15 and 29% of the farmers at the remining sites also used alternative pest management options. The farmers tended not to use fungicides in rice because they perceived that there were no disease problems (42–61%), but more farmers at Rizal (40%) compared to the other sites (17–26%) also used alternatives management practices (see below) to avoid using fungicides (L-R χ^2^ = 29.303, DF = 18, *p* = 0.045, Table 2). The farmers that used no herbicides in their rice, used hand-weeding instead (L-R χ^2^ = 10.211, DF = 9, *p* = 0.333, Table 2).

### 3.6. Other Activities in Rice Fields

More farmers at Iloilo (75%) rotated their rice with other crops compared to the farmers at Rizal (50.00%) and Bukidnon (36.36%), with only few farmers at Laguna (10.71%) rotating their rice crop (χ^2^ = 64.058, *p* ≤ 0.001). The farmers mainly rotated rice with field crops (21.51%—mainly maize (*Zea mays* L.)) and legumes (56.40%, mainly mung bean (*Vigna radiata* (L.) R. Wilczek)). The farmers’ reasons to rotate crops differed between regions, i.e., at Laguna, the farmers mainly rotated their rice to optimize the available space between successive rice crops (75.00%), whereas those in the other regions mainly rotated rice due to water shortages (Rizal = 54.84%, Iloilo = 30.77%, Bukidnon = 34.38%) and for pest management (3–16% of farmers) (L-R χ^2^ = 24.884, DF = 12, *p* = 0.015). More farmers at Iloilo (75.00%) allowed their rice to ratoon compared to farmers in the other regions (44.64–50.00%) (χ^2^ = 19.954, DF = 3, *p* ≤ 0.001).

Many farmers used their rice fields to also raise ducks: more farmers at Iloilo (44.05%) and Rizal (30.36%) raised ducks than at Bukidnon (24.14%) or Laguna (14.29%) (χ^2^ = 16.054, DF = 3, *p* ≤ 0.001). The farmers (66–79%: χ^2^ = 3.373, DF = 3, *p* = 0.338) also hand-picked snails as food, and 75–89% of them also wild-harvested other rice field animals as food (χ^2^ = 7.092, DF = 3, *p* = 0.252). The main animals harvested were frogs (61 farmers), fish and eels (22 farmers), and rats (18 farmers). Other animals from the rice fields that were consumed included snakes and turtles, birds and bird eggs, crawfish, and mole crickets (*Gryllotalpa* sp.).

### 3.7. Vegetable Production on Rice Farms

Over 80% of the farmers at Rizal, Iloilo, and Bukidnon grew vegetable or fruit crops on their farms, whereas only 46% of the farmers at Laguna grew vegetables or fruits (χ^2^ = 47.010, DF = 3, *p* ≤ 0.001). The farmers produced a diversity of other crops including legumes, gourds, squashes and melons, green and root vegetables, as well as fruit trees (Table 3). Further details of the plants grown on the farms are presented in Appendix A. 

In general, the farmers at Iloilo tended to grow more squashes and melons than the farmers at Laguna, Rizal, and Bukidnon (L-R χ^2^ = 18.009, *p* ≤ 0.001); meanwhile, the farmers at Iloilo and Bukidnon tended to grow more root vegetables (L-R χ^2^ = 16.564, *p* ≤ 0.001) and fruit crops (L-R χ^2^ = 13.444, *p* = 0.004) than those on Luzon. The farmers mainly produced fruits for home use (L-R χ^2^ = 24.109, *p* ≤ 0.001). Okra (*Abelmoschus esculentus* (L.) Moench) was also mainly grown for home use (L-R χ^2^ = 3.684, *p* = 0.055), whereas squashes and melons were mainly sold at the market (L-R χ^2^ = 5.585, *p* = 0.020). There were no significant regional or end-use trends for the remaining crops (Table 3).

### 3.8. Pest Management in Vegetable Crops

The farmers at Laguna (home use: 1.25 ± 0.37; market: 6.29 ± 1.36) and Rizal (home use: 5.24 ± 1.68; market: 5.61 ± 1.22) made more pesticide applications to their vegetables than the farmers at Iloilo (home use: 2.04 ± 0.72; market: 3.36 ± 0.47) and Bukidnon (home use: 2.06 ± 0.33; market: 2.48 ± 0.52) (F_3,238_ = 6.345, *p* ≤ 0.001). The farmers applied more pesticides to crops produced for the market (F_1,238_ = 15.503, *p* ≤ 0.001); however, there was a significant interaction effect, because the farmers at Rizal made similar numbers of pesticide applications to crops for home use and the market (F_3,238_ = 2.894, *p* = 0.036) (Figure 2). 

The farmers made fewer applications of insecticides and fungicides to vegetables produced for home use; they tended to make few herbicide applications to their vegetables, but only the farmers at Laguna made fewer applications to vegetables for home consumption—resulting in a significant interaction effect (Figure 2). More crops grown for home use were fungicide- or insecticide-free, but fewer farmers at Rizal produced fungicide-free or insecticide-free crops for home use (Figure 2B,D). The main insecticides used on vegetables were cypermethrin, lambda cyhalothrin, chlorpyrifos, methomyl, and malathion; the main herbicides were glyphosate, 2-4D, benzobicyclon, fenoxaprop-p-ethyl, and paraquat; and the main fungicides were benomyl, chlorothalonil, ethylene bisdithiocarbamate, strobilurin, and copper hydroxide. 

Overall, between 9% (Rizal) and 59% (Bukidnon) of the farmers produced crops for home use that were entirely pesticide-free, whereas 27% and 28% of the farmers at Iloilo and Bukidnon, respectively, produced pesticide-free crops for the market. None of the farmers at Laguna or Rizal produced pesticide-free crops for the market (site: χ^2^ = 16.064, DF = 3, *p* = 0.001; use: χ^2^ = 19.489, DF = 1, *p* ≤ 0.001). When asked specifically whether they applied pesticides to crops grown for home use, 25% (Rizal) to 70% (Iloilo) of the farmers said they avoided chemical pesticides on farm products for home use (χ^2^ = 10.745, *p* = 0.013; 180 valid responses).

### 3.9. Farmers’ Reasons to Avoid Pesticides in Vegetables

Farmers’ reasons for avoiding insecticides on vegetables were similar across sites and for both end uses (i.e., home use and market, Table 4), with farmers mainly avoiding insecticides because they used other control methods and for health reasons. The farmers avoided herbicides on vegetables mainly for health reasons at Iloilo and Bukidnon, but the farmers at Laguna felt there was no need for herbicides in their vegetable crops (Table 4). Farmers growing vegetables for the market tended to avoid herbicides for environmental reasons, including to avoid phytotoxic effects on their vegetables or on the main rice crop (Table 4). The farmers at Iloilo and Bukidnon were more likely to avoid fungicide applications for health reasons (Table 4).

### 3.10. Alternatives to Chemical Pesticides in Rice and Vegetables 

The farmers avoided insecticides by applying a range of alternative practices to reduce pest numbers or increase the abundance of natural enemies. Alternative practices differed between sites (L-R χ^2^ = 18.527, DF = 9, *p* = 0.030) and were different for rice and vegetable crops (L-R χ^2^ = 19.297, DF = 3, *p* ≤ 0.001, Table 5). More insecticide-free farmers at Bukidnon used *Trichogramma* cards or other biological control agents (χ^2^ ≤ 0.05) or sprayed botanical insecticides and organic ‘concoctions’ (farmers’ own term) on their rice and vegetables compared to farmers at the other sites (χ^2^ ≤ 0.05). Fewer farmers at Laguna planted flowers on their bunds compared to the other sites (χ^2^ ≤ 0.05) (Table 5). 

The farmers who did not use herbicides conducted manual weeding for both rice and vegetables; they also used cultural control methods such as flooding the paddies, plowing, and puddling (mainly for rice). Over 90% of the farmers mentioned manual weeding across sites and crops. Fungicide-free rice and vegetable crops were largely managed by manually removing diseased plants or plant parts or by applying organic fungicides and growth stimulants (Table 5). The use of non-chemical disease controls was similar across sites (L-R χ^2^ = 10.219, DF = 6, *p* = 0.081) and crops (L-R χ^2^ = 3.094, DF = 2, *p* ≤ 0.001). 

A number of the insecticide-free rice farmers grew vegetables on their bunds as a pest management action. The farmers tended to apply more botanical insecticides, and more farmers applied cultural controls on their vegetable crops than on their rice. In contrast, more farmers used biological control for their rice than for their vegetables. Some of the botanical insecticides and ‘concoctions’ are indicated in Table 5.

### 3.11. Predictors of Pesticide Applications in Rice

Insecticide use and overall pesticide use had the lowest AICc values in the DistLMs and explained 22% and 19% of all the variability in the data, respectively (Table 6). The ‘arthropod recognition score’, ‘region’, ‘flowers on bunds’, and ‘raising ducks’ were among the top variables contributing to variance in insecticide and pesticide use, although with different weights: the ‘arthropod recognition score’ explained 5.7% of the variance in insecticide use, whereas ‘flowers on bunds’ explained 6.3% of the variance in pesticide use (Table 6). ‘Flowers on bunds’ appeared in four of the five models (insecticide use, herbicide use, molluscicide use, and pesticide use). Two predictor variables, ‘arthropod recognition score’ and ‘education’, appeared in three models (insecticide use, fungicide use, and pesticide use, as shown in Table 6).

## 4. Discussion

Our study aimed to describe rice farm diversification and pest management practices in four regions that were selected by the Philippine DA to promote ecological engineering. We also assessed whether farm diversification and related activities affected the farmers’ use of chemical pesticides, particularly in rice. The farmers we interviewed were typical farmers of South and Southeast Asia that produce rice on small landholdings of < 2 ha [22,23,32,33,38]. The responses to our questionnaire revealed that the farmers in each of the four regions, although mainly dedicated to rice production, also produced a variety of other crops and livestock. The rice production landscape therefore consisted of numerous small farms that variously included patches of native vegetation, regenerating bushland, dryland crops, and home gardens [32,33]. The farmers reported over 60 plant species (Appendix A) that they grew in the proximity of their rice fields—and sometimes on their rice bunds. These dryland crops were mainly grown as a source of supplementary income, but also for home consumption. However, several farmers also grew plants on their bunds or near their rice crop to repel pests or to attract natural enemies (Table 5). The planting of flowers on rice bunds was among the strongest predictors (Table 6) of a relatively low pesticide use by the farmers in their main rice crop. Many farmers also prepared botanical extracts or purchased commercially available growth stimulants to manage insect pests and diseases in rice and, more frequently, to manage pests in their vegetables (Table 5). Some farmers also used their rice paddies to raise ducks and harvest wild animals such as frogs, fish, eels, and, sometimes, apple snails as food. Our results indicated that farmers that raised ducks (Table 6) were less likely to apply insecticides or molluscicides in their paddies, further supporting the idea that diversification reduced pesticide use and that the co-production of rice and aquatic or semiaquatic livestock can be an effective strategy to avoid pesticides on smallholding farms [22,32,38]. 

In the following sections, we will discuss some of the sustainable farming practices used by the farmers and their implications for promoting ecological engineering in the study regions. We will also discuss the motivations of the farmers to reduce pesticide use and the importance of alternative management systems to help farmers break the lock-in with respect to pesticide use.

### 4.1. Pesticide Use by the Farmers

The farmers we interviewed made, on average, about one application of herbicide, fungicide, insecticide, and molluscicide to their rice crop each season. Insecticide and molluscicide use were more variable, with the farmers in Rizal and Iloilo tending to perform at least a second insecticide application and use more molluscicide than the farmers in the other regions (Figure 1). Although these application rates are relatively low compared to those in other tropical rice production regions [19,22,39], it is notable that many farmers applied relatively hazardous chemicals to their rice and vegetables, including cypermethrin, chlorpyrifos, and lambda-cyhalothrin. Some farmers also still used paraquat to clear fields for vegetables. Each of these chemicals has been associated with severe environmental or health effects. For example, cypermethrin was shown to induce outbreaks of planthoppers in rice, partly by reducing the rice plant’s defenses against these herbivores [16,18,20], and chlorpyrifos and paraquat were associated with acute and chronic effects on human health [40,41,42]. Since many of the farmers reported the trade names and not the active ingredients, we noted that a large proportion (ca. 37% of the insecticides) of the chemicals used by the farmers were formulated and packaged in the Philippines as generic, off-patent pesticide products. The use of such problematic insecticides in rice, which is known as a highly resilient crop [15], exemplifies why alternatives to pesticide-based crop management are required.

### 4.2. Tailoring Vegetation Strips to Meet Farmers’ Needs

Smallholder farmers are often more amenable to adopting sustainable farming methods, particularly if these are associated with diversifying farm produce with supplementary goods for home use or as a source of extra income [32,33]. Most of the farmers we interviewed maintained relatively small vegetable patches (Appendix A) of fruits, vegetables, and herbs (Table 3, Appendix A). Some farmers (47%) also rotated their rice with upland crops such as mung bean (at Iloilo and Bukidnon), soybean (*Glycine max* (L.) Merr.) (Rizal), or melons (*Cucumis melo* L.) (Rizal) to take advantage of the market prices, improve soil conditions, or because of water shortages during the dry season—particularly on rainfed farms. A number of farmers (64%) used their bunds to grow vegetables and/or flowers. The flowers were mainly used as repellents or to promote natural enemies for rice pest management; several farmers mentioned that they integrated flowers or herbs such as cosmos (*Cosmos* spp.), marigolds (*Tagetes* spp.), and lemongrass (*Cymbopogon* sp.) with vegetables on their bunds as a means of reducing pest attacks on vegetables. The general awareness among farmers of the benefits of diversifying their farms and of growing vegetables and flowers on their bunds may be due to the considerable investment by the Philippine Rice Department (PhilRice) in promoting Palayamanan and/or similar sustainability programs. The Palayamanan Program, established in the early 2000s, promotes farm diversification by integrating fruit and vegetable production, livestock rearing, and aquaculture with rice production. The program further promotes a high-level of communication and cooperation between farmers, extension technicians, researchers, and other relevant stakeholders. However, the promotion of Palayamanan has been uneven across Philippine regions and municipalities [43], which may have contributed to the regional variability in farmers’ adoption of sustainable practices and in their ecological knowledge, as observed in our study (Table 1).

Research on the role of vegetables and flowers grown on bunds for rice pest management indicated that certain plants, such as sesame (*Sesamum indicum* L.) and mung bean, are used by predators such as *C. lividipennis* as well as the adults of parasitic wasps (e.g., *Anagrus* spp., *Cotesia chilonis* (Munakata), *Trichogramma chilonis* Ishii, *T. japonicum*, etc.) that kill the eggs of planthoppers, stemborers, and rice bugs [19,27,28,44]. These plants are thought to provide alternative prey items that sustain these natural enemies during periods of low prey abundance in the rice crop or, alternatively, produce nectar that sustains the free-living life stages of parasitic wasps, thereby increasing their longevity, fecundity, and impact on pests [26]. Furthermore, by planting flowers or vegetables on their rice bunds, the farmers provide a habitat for spiders and other predators that move between the bunds and the rice crops [44]. Some of the flowering plants and herbs that the farmers mentioned during the survey are known for their ability to reduce pest damage to rice and vegetables in small field plots [45,46,47]. However, in general, there is little information about which flowering plants might reduce rice pests at larger scales; indeed, some of the more detailed field experiments, conducted at relevant scales (fields or farms), suggest that the effects of planted bunds may extend only over a short distance into the rice fields [19,44]. This suggests that intercropping repellent flowers or herbs with vegetables on the bunds will at least protect the vegetables against damage, with beneficial effects that may spill over to the rice crop. However, the choice of bund vegetation must also consider the possible exacerbation of pest damage to rice [31,44].

The farmers in the four regions ranked their pest problems differently; notably, the farmers in Bukidnon were most concerned about stemborers, whereas those in Laguna and Rizal were concerned with rats and rice bugs. Damage from the white stemborer (*Scirpophaga innonata* (Walker)) can be severe in Bukidnon and other parts of Mindanao [48]. Many of the farmers (42%) that we interviewed in Bukidnon used *Trichogramma* egg cards, supplied through the Regional Crop Protection Centre (RCPC) in Malaybalay, to control stemborer damage. Most of these farmers (47%) avoided using insecticides altogether in rice. Where stemborers are problematic, farmers can grow trap plants such as vetiver grass (*Vetiveria zizanioides* (L.) Roberty) or Sudan grass (*Sorghum × drummondii* (Nees ex Steud.) Millsp. and Chase) [49,50]. Ecological engineering with nectar-producing plants—such as mung bean—is also compatible with biological control because it sustains the adults of *Trichogramma* wasps [19,28,30]. For the farmers at Laguna and Rizal, concerned about rats and rice bugs, linear strips of vegetation might be better avoided. Based on observations of the damage to rice on overgrown rice bunds from rodents, Horgan et al., 2017 [30] suggested that functional plants should be grown in patches separated by bare (mowed) bunds, thereby reducing rat foraging. Overall, the surveys revealed that many of the farmers desired to use alternative pest management approaches, including planting flowers and vegetables on rice bunds, but that they still lacked an objective knowledge of the available alternatives and of the best options for their particular needs.

### 4.3. Farmers’ Use of ‘Concoctions’ and Other Pest Management Options 

In contrast to the scant evidence for the repellent effects of planted flowers in field crops, there is growing evidence that homemade and commercialized botanical extracts, including those used by the farmers in our study, can have significant positive effects in crop protection, including in rice protection [51,52,53,54,55]. Many of the farmers we interviewed avoided using insecticides and fungicides by applying ‘concoctions’, herbal sprays, and foliar fertilizers. The substances used by the farmers included indigenous microorganisms, vesicular arbuscular mycorrhizae root inoculant, oriental herbal nutrient, fermented fruit juice, fermented plant juice, and a diversity of botanical extracts. A small number of farmers also purchased products that were marketed as organic conditioners or growth stimulants. These products were predominantly used to protect vegetables, although a small number of farmers (7% = 21 farmers) did apply these products to their rice crops—sometimes together with chemical pesticides (29% = 6 farmers). The microorganisms, root inoculants, fermented fruit juices, and herbal nutrients are primarily designed to revive soil nutrients (such as zinc and potassium) and improve soil fertility, while also enhancing the natural defenses of vegetable crops, particularly against nematodes and fungal diseases [56,57,58,59]. Botanical extracts are generally prepared as chopped and diluted compounds that are sprayed directly on plants or pests. These can contain extracts of lemongrass, chili (*Capsicum* spp.), garlic (*Allium sativum* L.), ginger (*Zingiber officinale* Roscoe), madre de cacao (*Gliricidia sepium* (Jacq.) Steud), or other plants that have shown negative effects on insect pests [51,52,60,61]. Some of these extracts were shown to contain insecticidal compounds that control herbivorous pests in the field [51]. The Philippine DA Agricultural Training Institute developed guidelines for farmers to prepare these alternatives at home in an attempt to promote their wider use [62]. Based on our results, these alternatives played a key role in weaning farmers away from using chemical pesticides. 

Because of the low financial returns from research into developing homemade botanical extracts for pest management, these technologies remain understudied [51,52]. However, their use reduces profitability losses associated with prophylactic chemical pesticide applications and the resulting environmental contamination [18]. Nevertheless, botanical extracts can still be damaging to human and livestock health (causing burns and irritation), can be phytotoxic to rice plants and other crops, and can interfere with rice ecosystem services—including regulatory ecosystem services [63,64]. Despite these issues, the careful consideration of botanical products represents a novel opportunity to expand on the practices of ecological engineering and further reduce pesticide use. For example, many farmers in Vietnam who established flower and vegetable strips as an ecological engineering practice failed to reduce pesticide inputs, and some of them applied insecticides to their bund-grown vegetables [22,31]. By encouraging farmers to grow plants that could be used to prepare botanical pesticides, farmers might be encouraged to further avoid chemical pesticides. In effect, this would provide a further benefit from the vegetation, helping promote ecological engineering, and would likely increase the adoption of this method by farmers (Figure 3).

### 4.4. Farmers’ Motivations to Avoid Pesticide Use

When we asked the farmers why they avoided using pesticides, the most common answer was generally that they applied alternative pest management methods (as discussed above). Although this does not explain their motivations, it does, nevertheless, indicate the importance of available alternatives in reducing pesticide inputs. When the interviewers tried to understand the motivations behind using alternatives, the farmers appeared to be predominantly concerned about the possible health impacts of using pesticides in rice and about the environmental impacts of pesticides, including their phytotoxic effects on adjacent rice or vegetables, the depletion of natural enemy abundance, and a possible reduction in soil fertility (Table 2). We also found a relatively strong negative relation between the farmers’ ability to recognize arthropods as beneficial or pestiferous and their use of insecticides (Table 6). Because of the choice of arthropods used in our evaluations, our simple index also indicated farmers’ leniency toward rice field arthropods: the farmers that were less inclined to categorize insects as pests made fewer pesticide applications.

The farmers were generally open in explaining that they avoided pesticides in rice and vegetables that they grew for home consumption. Furthermore, the farmers reported significantly lower insecticide and fungicide inputs in vegetables grown for home use (Figure 2). A few farmers indicated that their avoidance of chemical pesticides in rice and vegetables for home use was for health reasons (presumably, they were concerned about contaminated food). However, it is possible that the farmers also avoided pesticide on crops for home use because these did not generate any income and therefore did not justify added input costs [65,66]. Furthermore, farmers may be driven to achieve maximum protection when products are destined for the market (i.e., the farmers themselves were tolerant of damaged fruits and vegetables, but the retailers were not) [67]. Therefore, it is important to note that the motivations are not necessarily sinister (i.e., the farmers were concerned about the health of their own families but not about the health of others) but are more likely a feature of cost–benefit ratios and market demands. For example, farmers at Rizal made similar pesticide applications to vegetables for home use and for the market (Figure 2); in addition, the farmers often applied botanical extracts to vegetables destined for the market, thereby avoiding pesticide use. Other mechanisms to reduce insecticide use in vegetables might therefore include more frequent and consequential tests of pesticide residues at markets and greater traceability of market produce [68].

A number of studies indicated that farmers that use their rice paddies for fish, shrimp, or crab farming, particularly if using integrated approaches, will avoid using pesticides for their rice [22,38,69]. This is mainly to avoid possible fish kills, but also because fish consume rice pests [70,71]; 33% of the farmers we interviewed produced livestock on their farms. This mainly included cattle, goats, pigs, and ducks. Whereas mammal livestock are mainly restricted to dryland areas on farms, ducks forage in rice fields for weeds, snails, and insects. We found that raising ducks (29% = 83 farmers) was associated with reduced pesticide use, contributing 1.6% variance to insecticide use and 3.8% variance to molluscicide use (Table 6). Such an association may be expected for molluscicides because the ducks consume large numbers of apple snails and are frequently proposed as an alternative to molluscicides [72,73]. However, ducks also eat a range of insects and consume weeds [74,75], and the farmers may have been encouraged to avoid insecticide use to ensure that their ducks foraged in fields that were free of insecticide contamination or that maintained sufficient populations of invertebrates for the ducks to forage. Many of the rice farmers also wild-harvested frogs, fish, snails, and eels as supplementary food. We found no relation between these activities and the use of any pesticides. A relationship might have been expected if the farmers avoided consuming contaminated, wild-caught foods. However, we did not ask the farmers how often they consumed wild-caught animals and suggest that such activities were largely opportunistic and not frequent.

## 5. Conclusions

The rice farmers we interviewed frequently produced other crops and livestock on their farms. These included a diversity of plant and animal species. Furthermore, the farmers occasionally wild-harvested a range of animals including fish, frogs, and snails for home consumption. These approaches to intensifying their farm productivity while at the same time diversifying their products and optimizing the environmental capital of their farms contributed to reduced pesticide applications to the rice crop. In cases where farmers produce crops as supplementary food, these activities can also reduce the overall pesticide inputs in farms, in part because farmers view diversified farms as healthier environments for natural enemies and because they tend to avoid pesticides on produce for home use by applying concoctions and other alternative management practices, including the use of biological control agents. The survey results indicate that farmers that practice ecological engineering might therefore be encouraged to explore the use of concoctions that they can derive from the planted rice bunds, thereby increasing the benefits of the planting of vegetation or flower strips. The results of this case study have implications for rice production systems not only in the Philippines but also in other parts of South and Southeast Asia where farmers operate small, diversified farms.

## Figures and Tables

**Figure 1 insects-14-00778-f001:**
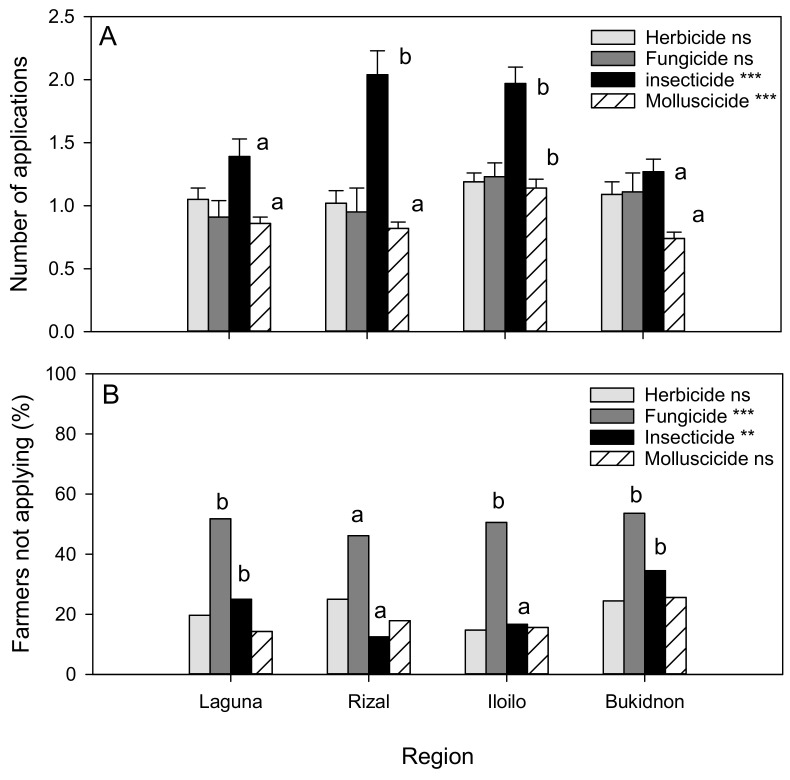
Farmers’ use of agrochemicals for pest management in rice. The average number of pesticide applications made by the farmers (**A**) and the proportion of farmers making no applications (**B**) are indicated for each surveyed region. The effects of the region on pesticide use are indicated as ns = *p* ≥ 0.05, ** = *p* ≤ 0.01, and *** = *p* ≤ 0.001 (degrees of freedom: **A** = 3,294; **B** = 3); lowercase letters indicate homogenous region groups (**A**: Tukey = 0.05; **B**: χ^2^ = 0.05). Standard errors are indicated in (**A**). The numbers of applications were ranked before the analyses.

**Figure 2 insects-14-00778-f002:**
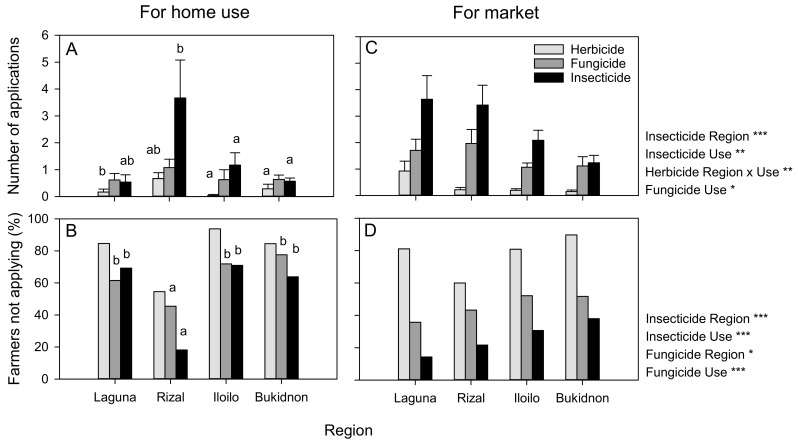
Farmers’ use of agrochemicals for pest management in vegetables. The average number of pesticide applications made by the farmers (**A**,**C**) and the proportion of farmers not making applications (**B**,**D**) are indicated for crops grown for home use (**A**,**B**) and for the market (**C**,**D**) in each surveyed region. The effects of the region on pesticide use are indicated as * = *p* ≤ 0.05, ** = *p* ≤ 0.01, and *** = *p* ≤ 0.001 (degrees of freedom: **A** = 3,240 for region; 1,240 for end use, and 5250 for interactions; **B** = 3 for region, 1 for end use); lowercase letters indicate homogenous region groups based on pesticide applications during production for both home use and the market (**A**,**C**: Tukey = 0.05; **B**,**D**: χ^2^ = 0.05). Standard errors are indicated in (**A**,**C**). The numbers of applications were ranked before the analyses.

**Figure 3 insects-14-00778-f003:**
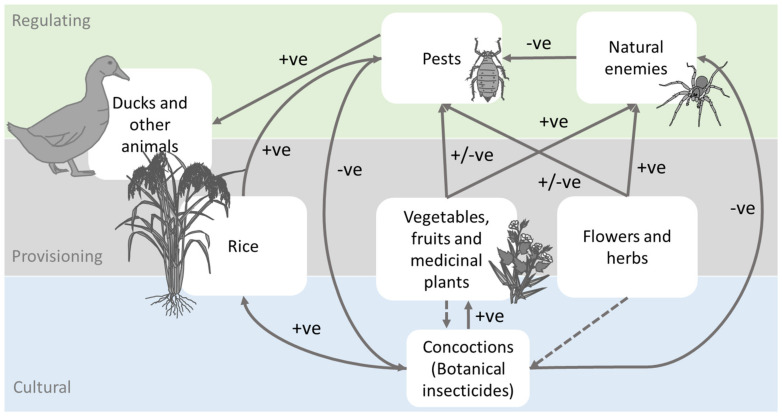
Links between components of sustainable rice production for farmsteads indicating the roles of flower and vegetable strips, raising ducks, and homemade botanical concoctions in pest management. Component categories of ecosystem services are also indicated. Note that the home production of ‘concoctions’ is included as a cultural ecosystem service linked to vegetable and flower strips that diminishes the cultural prominence of chemical pesticides. As such, homemade botanicals could encourage farmers to further adopt ecological engineering. Arrows indicate the direction of material and energy flows and are designated as positive (+ve) or negative (−ve). Note that homemade concoctions can have negative impacts on natural enemies that are less severe than those of chemical pesticides.

**Table 1 insects-14-00778-t001:** Farmers’ recognition of rice field arthropods (shown to the farmers in photographs) as beneficial or pestiferous.

Species	Status	Region ^1,2^				χ^2 2^
		Laguna	Rizal	Iloilo	Bukidnon	
Dragonfly	Beneficial	1.79	1.82	4.17	4.55	1.391 ns
Bee	Beneficial	16.07	19.64	23.96	13.48	3.647 ns
Planthopper	Pestiferous	85.71	91.07	85.42	86.52	1.120 ns
Wasp	Beneficial	12.50 ^a^	46.43 ^b^	37.50 ^b^	26.97 ^a^	17.654 ***
Rice bug	Pestiferous	96.43	90.91	94.74	90.91	2.458 ns
Mirid	Beneficial	56.00	74.00	56.25	53.41	6.214 ns
Ladybeetle	Beneficial	41.51 ^a^	80.00 ^b^	81.25 ^b^	40.91 ^a^	48.534 ***
Egg parasitoid	Beneficial	37.25 ^a^	62.00 ^b^	53.13 ^ab^	30.68 ^a^	16.929 ***

^1^: Numbers are the percentage of farmers from each region that identified the insects as pests. ^2^: The effects of the region on the farmers’ responses for each species are indicated as ns = *p* ≥ 0.05, and *** = *p* ≤ 0.001 (degrees of freedom = 3); lowercase letters indicate homogenous region groups (χ^2^ ≤ 0.05).

**Table 2 insects-14-00778-t002:** Reasons why the farmers avoided using chemical pesticides in rice in the four regions.

Reasons for Avoiding Pesticides	Region ^1,2^			
	Laguna	Rizal	Iloilo	Bukidnon
Insecticides				
a Use alternative methods or practice organic farming	23.08 ^a^	28.57 ^a^	15.00 ^a^	68.97 ^b^
b No need to apply	30.77 ^b^	28.57 ^b^	35.00 ^b^	10.34 ^a^
c Believe that natural enemies protect the crop	30.77 ^b^	42.86 ^b^	20.00 ^b^	10.34 ^a^
d To avoid health-related impacts	7.69	0.00	20.00	6.90
e Based on advice from the DA or others	7.69	0.00	10.00	0.00
f To avoid/reduce production costs	0.00	0.00	0.00	3.45
Herbicides				
a Use alternative methods or practice organic farming	100.00	90.91	90.00	96.15
b To avoid/reduce production costs	0.00	9.09	0.00	0.00
c To protect/conserve natural enemies that protect the crop	0.00	0.00	10.00	3.85
Fungicides				
a No need to apply	61.11	45.45	57.14	41.94
b Use alternative methods or practice organic farming	16.67 ^a^	40.91 ^b^	19.05 ^a^	25.81 ^a^
c To avoid health-related impacts	0.00	0.00	14.29	25.81
d To avoid/reduce production costs	0.00	4.55	9.52	6.45
e Based on advice from the DA or others	11.11	9.09	0.00	0.00
f Believe that natural enemies protect the crop	11.11	0.00	0.00	0.00

^1^: Numbers indicate the percentages of farmers from each region that mentioned the corresponding reasons. ^2^: Lowercase letters indicate homogenous groups based on tests of partial independence (χ^2^ ≤ 0.05).

**Table 3 insects-14-00778-t003:** Fruit and vegetable production on the rice farms.

Crops Produced	Home Use (% of Farmers) ^1^				Market (% of Farmers) ^1^				L-R χ^2^ Region ^2^	L-R χ^2^ End-Use ^2^
	Laguna	Rizal	Iloilo	Bukidnon	Laguna	Rizal	Iloilo	Bukidnon		
Okra	66.67	63.64	48.39	66.67	35.71	62.16	44.68	44.44	4.497 ns	3.684 *
Solanaceae	50.00	45.45	35.48	52.54	64.29	62.16	46.81	59.26	4.570 ns	1.604 ns
Legumes	25.00	18.18	32.26	29.82	21.43	21.62	23.40	44.44	3.356 ns	0.077 ns
Greens	25.00	27.27	19.35	17.54	21.43	13.51	31.91	18.52	2.738 ns	0.234 ns
Gourds	8.33	18.18	12.90	24.56	35.71	23.68	14.89	22.22	2.814 ns	0.261 ns
Squashes and melons	8.33 ^a^	27.27 ^a^	25.81 ^b^	5.26 ^a^	21.43 ^a^	16.22 ^a^	40.43 ^b^	14.81 ^a^	18.009 ***	5.385 *
Root crops	0.00 ^a^	9.09 ^a^	19.35 ^b^	28.07 ^b^	0.00 ^a^	2.70 ^a^	19.15 ^b^	22.22 ^b^	16.564 ***	2.674 ns
Fruit trees	0.00 ^a^	0.00 a	32.26 ^b^	28.07 ^b^	0.00 ^a^	2.70 ^a^	2.13 ^b^	3.70 ^b^	13.444 ***	24.109 ***
Field crops	16.67	9.09	0.00	7.02	7.14	2.70	4.26	7.41	3.713 ns	0.256 ns

^1^: Percentages are only for farmers that grew crops other than rice; lowercase letters indicate homogenous region groups based on tests of partial independence (χ^2^ ≤ 0.05). ^2^: ns = *p* > 0.05, * = *p* ≤ 0.05, and *** = *p* ≤ 0.001. Degrees of freedom = 3 for regions and 1 for end uses; there were 236 valid responses. For further details of the plants produced on the farms, see Appendix A.

**Table 4 insects-14-00778-t004:** Farmers’ reasons for avoiding pesticides in vegetables.

Reasons for Pesticide-Free Vegetable Production	Home Use (% of Farmers) ^1^				Market (% of Farmers) ^1^				L-R χ^2^ -Region ^2^	L-R χ^2^ -End Use ^2^
	Laguna	Rizal	Iloilo	Bukidnon	Laguna	Rizal	Iloilo	Bukidnon		
Insecticides										
a. Use other controls	37.50	0.00	47.37	55.26	50.00	44.44	45.45	22.22	19.836 ns	3.344 ns
b. For health reasons	12.50	100.00	47.37	10.53	0.00	22.22	45.45	22.22		
c. To preserve natural enemies	25.00	0.00	0.00	10.53	0.00	33.33	0.00	44.44		
d. No need	12.50	0.00	5.26	21.05	50.00	0.00	9.09	11.11		
e. Advice from DA or others	12.50	0.00	0.00	2.63	0.00	0.00	0.00	0.00		
Herbicides										
a. Use other controls	55.56	83.33	62.96	55.56	50.00	80.00	76.32	50.00	50.770 ***	12.708 *
b. For health reasons	0.00 ^a^	0.00 ^a^	29.63 ^b^	44.44 ^b^	16.67 ^a^	0.00 ^a^	10.53 ^b^	36.36 ^b^		
c. Avoid environmental and phytotoxic effects	11.11	16.67	0.00	0.00	16.67	13.33	7.89	4.55		
d. No need	22.22 ^b^	0.00 ^a^	7.41 ^a^	0.00 ^a^	0.00 ^b^	6.67 ^a^	5.26 ^a^	4.55 ^a^		
e. To avoid added costs	11.11	0.00	0.00	0.00	16.67	0.00	0.00	4.55		
Fungicides										
a. Use other controls	14.29	80.00	31.82	67.57	40.00	50.00	44.44	64.29	22.652 ***	1.572 ns
b. No need	85.71 ^b^	20.00 ^a^	36.36 ^a^	16.22 ^a^	40.00 ^b^	45.00 ^a^	33.33 ^a^	21.43 ^a^		
c. For health reasons	0.00	0.00	31.82	16.22	20.00	5.00	22.22	7.14		
d. To protect soil organisms	0.00	0.00	0.00	0.00	0.00	0.00	0.00	7.14		

^1^: Numbers are percentages of farmers indicating each reason (only one answer was accepted per farmer); lowercase letters indicate homogenous region groups based on tests of partial independence (χ^2^ < 0.05). ^2^: ns = *p* > 0.05, * = *p* ≤ 0.05, and *** = *p* ≤ 0.001. Degrees of freedom = 3 for regions and 1 for end uses; there were 100, 181, and 36 valid responses for insecticides, herbicides, and fungicides, respectively.

**Table 5 insects-14-00778-t005:** Alternative, proactive control methods used by farmers avoiding insecticides and fungicides on rice and vegetables.

Alternatives to Pesticides	Rice Crop ^4^				Vegetables ^4^			
	Laguna	Rizal	Iloilo	Bukidnon	Laguna	Rizal	Iloilo	Bukidnon
Insecticides								
a. Botanicals and organic insecticides ^1^	15.79 ^a^	8.33 ^a^	7.69 ^a^	24.19 ^b^	60.00 ^a^	28.57 ^a^	55.00 ^a^	70.37 ^b^
b. Functional plants ^2^	15.79 ^a^	50.00 ^b^	42.31 ^b^	32.26 ^b^	20.00 ^a^	57.14 ^b^	20.00 ^b^	22.22 ^b^
c. Use biocontrol agents	10.53 ^a^	16.67 ^a^	3.85 ^a^	30.65 ^b^	0.00 ^a^	0.00 ^a^	0.00 ^a^	7.41 ^b^
d. Manual removal/cultural ^3^	0.00	8.33	0.00	3.23	20.00	14.29	25.00	0.00
e. Produce vegetables on bunds	57.89	16.67	46.15	9.68	na	na	na	na
Valid cases	19	12	26	62	5	7	20	27
Fungicides								
a. Manual removal/cultural ^3^	0.00	88.89	0.00	85.71	0.00	60.00	84.62	50.00
b. Botanicals and organic pesticides ^1^	0.00	11.11	75.00	0.00	81.82	40.00	0.00	50.00
Valid cases	3	9	4	7	11	15	13	38

^1^: Farmers used a variety of botanical insecticides and growth-promoting agents, including foliar organic insecticides, to control insects. These included home-made liquids with detergents and soaps, chili (*Capsicum* spp.), ginger (*Zingiber officinale* Roscoe), camphor (*Camphora* sp.), madre de cacao (*Gliricidia sepium* (Jacq.) Steud.), and lemongrass (*Cymbopogon* sp.) extracts, fermented fruit juices, vinegar, urine, ash, as well as a range of ‘herbal’ sprays. Farmers also applied commercially available organic pesticides and growth stimulants. The category ‘botanical and organic insecticides’ also includes rustic traps used against rice bug and baited with rotting shrimp or snails, sometimes treated with *Metarhizium*; *Trichogramma japonicum* Ashmead was the main biological control agent used—the cards were supplied by local DA officers. ^2^: Functional plants included cosmos (*Cosmos* spp.), marigolds (*Tagetes* spp.), and lemongrass that were mainly used as repellents for insect herbivores. ^3^: Cultural methods included synchronized planting, using certified seed, and adjusting plant spacing to allow plants to grow larger (mainly for rice). ^4^: Numbers are percentages of farmers (only one answer was accepted per farmer); lowercase letters indicate homogenous region groups based on tests of partial independence (χ^2^ ≤ 0.05); na = not applicable.

**Table 6 insects-14-00778-t006:** Distance-based linear models (DistLMs) summary, with predictor variables included in each of the models.

Dependent Variable (Number of Predictors)	Model Summary			Sequential Tests			
	AICc	R^2^	Predictor Variables	Pseudo-F	*p* Values	% Var. Expl ^1^	% Cum. Var ^2^
Insecticide use (10)	306.66	0.220	Arthropod recognition score	17,504	0.001	5.7	5.7
			Flowers on bunds	11.67	0.002	3.6	9.3
			Region	7295	0.008	4.3	13.6
			Raising ducks	5560	0.025	1.6	15.3
			Hunt snails	5905	0.011	1.7	17.0
			Age	5659	0.016	1.6	18.6
			Rotate rice	5390	0.024	1.5	20.1
			Farm visits	2149	0.145	0.6	20.7
			Grow vegetables	2322	0.135	0.6	21.4
			Education	2359	0.126	0.6	22.0
Herbicide use (5)	447.76	0.085	Flowers on bunds	10,759	0.003	3.6	3.6
			Rice area	6001	0.008	1.9	5.5
			Region	3627	0.041	1.2	6.7
			Grow vegetables	3188	0.048	1.0	7.7
			Raising ducks	2481	0.112	0.8	8.5
Fungicide use (7)	321.41	0.094	Region	7611	0.009	2.5	2.5
			Education	3756	0.018	1.2	3.8
			Rice area	4198	0.028	1.4	5.2
			Arthropod recognition score	3315	0.036	1.1	6.3
			Grow vegetables	3048	0.041	1.0	7.2
			Raising ducks	4638	0.049	1.5	8.7
			Gender	2216	0.145	0.7	9.4
Molluscicide use (5)	522.24	0.110	Raising ducks	11,454	0.001	3.8	3.8
			Region	13,852	0.001	4.4	8.2
			Flowers on bunds	4586	0.008	1.4	9.6
			Grow vegetables	2551	0.012	0.8	10.4
			Harvest ratoon	2067	0.151	0.6	11.0
Pesticide use (9)	210.37	0.192	Flowers on bunds	19,654	0.001	6.3	6.3
			Region	14,285	0.001	5.0	11.4
			Arthropod recognition score	5675	0.017	1.7	13.1
			Raising ducks	4627	0.021	1.4	14.5
			Education	3461	0.022	1.0	15.5
			Age	4299	0.032	1.3	16.8
			Grow vegetables	3731	0.047	1.1	17.8
			Gender	2273	0.065	0.7	18.5
			Hunt snails	2315	0.128	0.7	19.2

^1^: % Var. Expl = the percentage of variance in the model explained by each variable. ^2^: % Cum. Var = the cumulative percentage of variance in the model.

## Data Availability

The data presented in this study are available on reasonable request to the corresponding author.

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
