# Peer review of "Diversified Rice Farms with Vegetable Plots and Flower Strips Are Associated with Fewer Pesticide Applications in the Philippines"

_insects, 2023, doi:10.3390/insects14100778_

Round 1
Author Response
We thank the reviewer for his/her kind comments and helpful suggestions for the manuscript. We have made all changes as requested by the reviewer. These are indicated by track changes in the text and listed here in order:
Simple summary
The reviewer suggests that a sentence is needed at the beginning of the simple summary to outline what the study set out to achieve; The reviewer also suggests that the recommendations at the end of the simple summary need to be more closely related to the study outcomes and should indicate how the results might help further develop ecological engineering practices (see also reviewer’s comment referring to lines 32-33/and 50-52).
RESPONSE: We have now added a sentence to explain the purpose of the study and made subsequent changes to the existing text to avoid any repetitions: “This study examines the impact of farm diversification and other sustainability practices on pesticide use by rice farmers. Consequently, we interviewed Filipino farmers to assess their farm management practices.”
We also amended the text to bring the main results to the end of the summary and outline their contribution to possible improvements of existing ecological engineering practices: “We found that farmers with relatively high awareness of the beneficial insects that occurred on their farms, that raised ducks in their rice fields, or that planted flowers or vegetables on their rice bunds tended to make fewer pesticide applications to their rice crops. We recommend that diversifying rice farms with flower and vegetable strips could be combined with a series of other, environmentally friendly pest management options to enhance the outcomes of ecological engineering.”
L20 The reviewer asks for clarification of the term ‘farmers’ at the beginning of the simple summary RESPONSE: We have indicated that these were ‘smallholder rice farmers’ and have amended the text according to this description.
L26 The reviewer asked for clarity around ecological literacy in the simple summary.
RESPONSE: We have removed reference to the term in the summary but leave the information that farmers that recognized arthropods were less likely to use pesticides: “We found that farmers with relatively high awareness of the beneficial insects that occurred on their farms …….. tended to make fewer pesticide applications to their rice crops.”
Ecological literacy as a term has been reviewed throughout the text and replaced with other information or terms (see below). We agree that the term is too broad to refer to our original definition and have avoided the term altogether.
L28 The reviewer suggests that some examples of homemade botanical extracts should be included.
RESPONSE: We have now included a series of examples: “Many farmers produced botanical extracts of chili (Capsicum spp.), ginger (Zingiber officinale Roscoe), camphor (Camphora sp.) or lemongrass (Cymbopogon sp.) to control pests and diseases in these vegetables.”
Abstract
The reviewer suggests that there is not enough structure at the beginning of the abstract to frame the research and that research aims and questions should be clearly outlined in the abstract.
RESPONSE: We have paid attention to this observation by the reviewer during rewriting of the abstract. We have now clarified the meaning of ecological engineering and the main aims of the study in the first sentences of the abstract: “Ecological engineering is defined as the design of sustainable ecosystems for the benefit of both human society and the environment. In Southeast Asia, researchers have applied ecological engineering to restore regulatory services to rice ecosystems and thereby reduce herbivore-related yield losses and overall pesticide use. We conducted a survey of 302 rice farmers across four regions of the Philippines to assess their farm diversification practices and determine possible associations with reduced pesticide inputs.”
L34-36 The reviewer suggests that a definition of ecological engineering is required at the beginning of the abstract. We have now included the definition and changed the remaining text accordingly.
RESPONSE: A definition has now been included.
44-45 The reviewer asks whether reference to farmers ‘producing rice and vegetables for home use made fewer pesticide applications compared to those that produced crops for market’ is significant.
RESPONSE: We have deleted this text from the abstract because it requires more discussion and its significance to the overall study is not sufficiently clear (i.e., as indicated in the discussion, this related to farmers’ health concerns, but also to the lower investment in products produced for home use).
L46 The reviewer asks that the sentence referring to a percentage ‘variously applied’ should be rewritten.
RESPONSE: Our original sentence was cumbersome in an effort to avoid inclusion of a several related percentages. To avoid this, and in response to the reviewer’s comment, we have now removed the sentence from the text and replaced it with the following: “……. and many farmers applied botanical extracts, growth stimulators and insect traps to reduce damage to vegetables.”
L47-49 The reviewer asks if the result related to the multivariate analyses were significant.
RESPONSE: We have clarified that there were significant associations between the listed factors and both insecticide and overall pesticide use: “The planting of flowers on rice bunds, rearing ducks in the rice fields, and the farmers’ recognition of beneficial rice arthropods, were statistically significantly associated with lower pesticide (particularly insecticide) applications to rice.”
L50-52 The reviewer again questions our use of the term ‘ecological literacy’.
RESPONSE: We have amended the text throughout the paper to avoid use of the term ‘ecological illiteracy’
L50-52 The reviewer suggests that the recommendations/outcome as presented in the abstract are an improvement over those in the simple summary. We originally aimed to not repeat information in the summary and abstract, however, this may have led to poorer text in the simple summary; therefore, based on the reviewer’s suggestion, we have placed a rewritten version of this last sentence of the abstract into the simple summary (see above).
Keywords
The reviewer asked that we remove 2 or 3 keywords.
RESPONSE: We have now removed keywords, including those that were repeated from the title.
Introduction
The reviewer provided an alternative structure for the introduction including: 1) importance of rice globally; 2) requirements for high yields lead to increased pesticide use; 3) detrimental effects on natural enemies of pesticides; 4) increasing outbreaks of pests due to resurgence; 5) natural methods for pest reduction (farmers often go back to pesticides); 6) ecological engineering; 7) research questions and aims
RESPONSE: We have now restructured the introduction according to the reviewer’s suggestions.
L57 The reviewer indicates that the opening sentence in the introduction requires further expansion to be relevant to the subsequent text.
RESPONSE: We have now expanded information around rice production and food security in response to the reviewer’s suggestion: “Rapid population growth (with the global population estimated to exceed 9 billion by 2050), particularly in tropical Asia, has increased pressures on Asian farmers to intensify rice production [Godfrey et al 2010; Seck et al 2012]. In response, rice intensification practices, including the use of high-yielding rice varieties, increased mechanization and an increasing use of agrochemical inputs, have been promoted by national and international agricultural development institutes, often in partnership with the private sector [Godfrey et al 2010; Tillman et al 2011].”
L71 The reviewer asks that the authors define technological lock-in.
RESPONSE: WE have now defined technological lock-in and further expanded on the context of increasing pesticide use in Asian rice: “A high use of chemical pesticides at large scales, and the co-dependence of certain technologies (i.e., hybrid rice varieties and direct-seeded rice are associated with higher pesticide use than traditional varieties and establishment methods [Horgan and Crisol 2013; Horgan 2018]) has resulted in a technological lock-in to pesticides (i.e., the increasing use of pesticides by farmers reduces their willingness to adopt more environmentally-friendly pest and weed management options [Wagner et al 2016; Bakker et al 2020; Jacquet et al 2022]).”
L108 The reviewer asks that ‘ecological engineering’ be better defined and that some examples should be included.
RESPONSE: We have now defined ecological engineering and included a well-known example: “Ecological engineering is defined as the design of ecosystems using engineering principals to promote benefits for both human societies and the environment [Mitsch and Jørgensen 2003]. In crop production systems, ecological engineering often relies on the use of functional plants (e.g., trap plants, repellent plants, or plants that provide alternative food sources for natural enemies) to increase the diversity and abundance of predatory arthropods. For example, the use of Lobularia maritima (L.) Desv. as a selective food plant for Trichogramma carverae Oatman & Pinto improves the biological control of Epiphyas postvittana (Walker) in Australian vineyards [Began et al 2006].”
Methods
The reviewer suggests that the term ecological literacy should be changes and suggests that arthropod literacy might be used instead.
L210 The reviewer suggests that the Ecological Literacy Index should have a paragraph on its own to better indicate what the index actually covers.
RESPONSE: We have now placed this information in a single paragraph; we also removed all references to ‘ecological literacy’ and replaced the text with references to farmers’ abilities to recognize arthropods as beneficial or pestiferous and to the simple arthropod recognition index.
Discussion
The authors suggests that the discussion could be reduced, particularly in sections 4.2, 4.3.
RESPONSE: We have reduced the length of the discussion, particularly for the sections as indicated by the reviewer.
L743 The reviewer asks that the overall aims of the study should be reiterated at the beginning of the discussion.
RESPONSE: We have reiterated the aims at the beginning of the discussion as follows: ‘Our study aimed to describe rice farm diversification and pest management practices at four regions that were selected by the Philippines DA to promote ecological engineering. We also assessed whether farm diversification and related activities affected the farmers’ use of chemical pesticides, particularly in rice.’
Other
Please confirm that all necessary ethics permissions were given.
RESPONSE: We have included information regarding surveys and data protection at the end of the methods section relating to the questionnaire: “Prior to conducting the interviews, the interviewers informed each farmer about the objectives of the interview, how the data would be used, and how the data would be stored (including that farmers’ names would only be recorded to match pre- and post-field-day-interviews, after which the names would be deleted such that, the reported results could not be linked to individual farmers). Farmers were also advised that they were not obliged to answer any questions.”
We have also added an institutional Review Board Statement and an Informed Consent Statement after the text.
Reviewer 2 Report
The paper describes findings on Filipino farmers’ literacy on ecological engineering in rice fields and how this affects applications of pesticides. Based on these, they made some recommendations for sustainable rice pest management. The research was conducted through surveys in four provinces on three islands of the Philippines. Data were collected by interviewing over 300 farmers. Design is appropriate to me though I have never done such research before. It is well written. I believe that paper is publishable as it is.
Author Response
We thank the reviewer for his/her kind comments. There were no requested changes by the reviewer.